# Child mental health predictors among camp Tamil refugees: Utilizing linear and XGBOOST models

Muna Saleh[1], Elizabeth Amona[2], Miriam Kuttikat[1]*, Indranil Sahoo[2], David Chan[3], Jennifer Murphy[4], Kyeongmo Kim[1], Hannah George[5], Marianne Lund[1]

1 School of Social Work, Virginia Commonwealth University, Richmond, Virginia, United States of America, 2 Department of Statistical Sciences and Operations Research, Virginia Commonwealth University, Richmond, Virginia, United States of America, 3 Department of Applied Mathematics, Virginia Commonwealth University, Richmond, Virginia, United States of America, 4 School of Social Work, University of Texas at Arlington, Arlington, TX, United States of America, 5 Department of Anthropology, George Washington University, Washington, DC, United States of America

* Kuttikatm@vcu.edu

**Data Availability Statement:** The data underlying th results presented in the study are publically available from https://github.com/indranil09/Child-Mental-Health-Predictor." –

## Abstract

While the association between migration and deteriorated refugee mental health is well-documented, existing research overwhelmingly centers on adult populations, leaving a discernible gap in our understanding of the factors influencing mental health for forcibly displaced children. This focus is particularly noteworthy considering the estimated 43.3 million children who are forcibly displaced globally. Little is known regarding the association between family processes, parental and child wellbeing for this population. This study addresses these gaps by examining the relationship between parental mental health and child mental health among refugees experiencing transmigration. We conducted in-person structured survey interviews with 120 parent-adolescent dyads living in the Trichy refugee camp in Tamil Nadu, India. Descriptive, multivariate analysis (hierarchical regression), and Machine Learning Algorithm (XGBOOST) were conducted to determine the best predictors and their importance for child depressive symptoms. The results confirm parental mental health and child behavioral and emotional factors are significant predictors of child depressive symptoms. While our linear model did not reveal a statistically significant association between child mental health and family functioning, results from XGBOOST highlight the substantial importance of family functioning in contributing to child depressive symptoms. The study's findings amplify the critical need for mental health resources for both parents and children, as well as parenting interventions inside refugee camps.

## Background

By the end of 2022, the global count of forcibly displaced individuals exceeded 108 million [1], presenting a pronounced rise from previous years. This figure includes 43.3 million children, highlighting the disproportionate impact of forced displacement on youth. A majority of

**Funding:** This work was supported by National institute of Health Fogarty International [grant numberK01 TW 009648].

**Competing interests:** This does not alter our adherence to PLOS ONE policies on sharing data and materials.

displaced persons are temporarily hosted in refugee camps in Lower Middle-Income Countries (LMICs), which often struggle with meeting the crucial psychosocial and health needs of refugees. These camps are notable for their deprivations in vital resources and neglect from international communities, often resulting in precarious living conditions characterized by consistent food insecurities, inadequate shelters, limited access to healthcare, and an array of other psychosocial challenges.

The recent retrenchment of global refugee policies has further compounded the challenges refugees face, imposing a heavier burden on host LMICs who must grapple with providing humanitarian services in the context of reduced international support and the potentially destabilizing social and economic environments that may arise from an influx of refugees. As an example, despite a decline in global participation in resettlement in recent years, the number of refugees in need of resettlement continues to rise. In 2022, only approximately 58,000 refugees were resettled—a stark contrast to the 116,000 applicants for resettlement [2]. These trends further contribute to restrictive host policies that may contribute to the exacerbation of existing disparities in refugee mental and physical health [3,4]. Additionally, the often prolonged nature of displacement–reflected in the lengthy average period of time refugees live in camp settings [5]–requires stakeholders and researchers to examine the unique mental health effects of transmigration for camp refugees.

Prior to reaching refugee camps, refugees are exposed to an array of psychosocial challenges associated with forced displacement, including exposure to traumatic stress that contribute to later psychopathology [6,7]. Upon arrival in host settings, refugees are confronted with opportunities for further traumatization, stemming not only from substandard living conditions [8] but also from chronic discrimination and marginalization in their social environments [9], and exposure to violence [10,11]. These challenges encompass arbitrary detainments, lack of legal documentation, exclusion from the labor market, and hostile policies (e.g., forced encampment) designed to control and confine refugees [12]. These social difficulties are exacerbated by discordant ethnic and cultural differences between refugees and the dominant identities in the host context, despite the fact that a majority of refugees (69%) seek safety in neighboring, culturally similar countries [1].

Children experiencing forced displacement represent an especially vulnerable population and are exposed to a range of conflict-related traumas [13] that are detrimental to their development. Experiences such as the loss/separation of a caregiver and their homes [14,15] exposure to violence (e.g. witnessing murder), forced labor, hunger, physical harm, perilous living conditions, poverty resulting from diminished family resources, and limited access to quality education and healthcare [13,16,17] are among the most common traumatic experiences of young refugee children. Exposure to traumatic stressors has been previously linked with the emergence of psychopathology in youth, with earlier experiences proving particularly detrimental to a child's well-being [18]. Not only are refugee children susceptible for experiencing traumatic stress, but they also remain vulnerable to cumulative exposures to traumatic stress, a significant risk factor for depression among refugee children [19]. The risk for cumulative stress is notably concerning given its resistance to treatment, compared with single-event traumatic exposures.

In the context of forced migration, numerous research consistently reveals high elevated rates of common mental disorders in refugee children [15,17,20–25]. In a study among children camp refugees Nasıroğlu and colleagues [26] found a prevalence rate for PTSD of 43.4% and 27.9% for depression. In another study, camp refugee children were found to endorse a depression rate of 20% [27]. Yayan and colleagues [28] found the mean score of depression among children in refugee camps–assessed using the Children's Depression Inventory–to be 60.59, indicating high levels of depression. Socioemotional and behavioral problems are also

prevalent among camp refugee children, with more than half (52%) of camp refugee children endorsing abnormal levels of emotional problems [14]. The long-term effects of war traumas are also notable, with a majority of former child camp refugees reporting high levels of PTSD long after the removal of stressors [10,28,29].

Previous research has demonstrated that parents experiencing migration-associated duress and enduring protracted statelessness may be particularly prone to mental health deterioration, posing a threat to parental coping and functioning and contributing to the intergenerational transmission of trauma [30–37]. Epidemiological studies among adult camp refugees consistently reveal high rates of PTSD, with some estimates exceeding 80% [38,39]. Similarly, depression is high in this population with estimates that include 38% [40] and 47% [41]. Collectively, these risks elevate the likelihood of impeded parenting resiliency among displaced parents and the transfer of maladaptive coping models to their children [42], as well as insecure parent-child attachments [43]. Although limited, previous studies have found direct links between parental and child psychopathology specifically among camp refugees [34,43–45].

Young refugee children may be deprived of the critical early development necessary for healthy brain development, which relies on stable psychosocial circumstances [46,47]. Epidemiological studies provide support for the protective role of social support for forcibly displaced children; Oppedal and colleagues [19] have found that among refugee children with high levels of exposure to traumatic stress, high levels of perceived socioemotional support contributed to resilient outcomes. Similarly, among camp refugees, parental behavior contributes to children's socioemotional behavioral problems [43] and the link between parental psychopathology and child mental health may be mediated by family functioning [48,49]. Moreover, refugee camp children report that their traumatic experiences are often mediated and exacerbated by family functioning, and at times, poor family functioning among camp refugees may even contribute to traumatization, resulting in poor mental health [33]. Due to the challenges of conducting health research in refugee camps, there is a scarcity of studies exploring the physical and mental health effects experienced during the transmigration process, especially among children, and their associations with family functioning [25].

A dearth of research is dedicated to developing and appraising mental health interventions aimed at child/adolescent refugees. Most of these interventions target child refugees in camp settings and focus primarily on reducing psychological distress resulting from exposure to traumatic experiences. Far more limited are the evaluation of interventions targeted at improving positive outcomes, such as resilience and well-being [50]. However, there have been more recent efforts to design interventions to improve functioning and socio-behavioral outcomes [50,51], which in turn, may prevent the development of secondary distress experienced in displacement. Notably, there are only a few studies evaluating the effectiveness of preventative interventions [50,52], and even less focus on refugees in camp settings that target family dynamics. In a systematic review on family-based interventions, Gillespie and colleagues [53] identified just four studies examining families living in camps or urban settings. Outcomes in those studies were varied, indicating the need for further research to enhance our understanding of family dynamics among refugees in camp environments.

## Present study

Building upon the research gaps identified above, our study investigates how parental transmigration stress influences child depressive symptoms and, more specifically, how displaced families navigate these challenging circumstances. The study is guided by the following research questions: (1) What factors best predict child depressive symptoms, their current levels of adjustment, and family functioning?; (2) Is there an association between child mental health

and parental mental health among refugees experiencing transmigration?; and (3) What are the most relevant factors in assessing the mental health of forcibly displaced children in camp settings? In addressing these research questions, we elucidate the relationships between parental mental health, child well-being, daily stress, and family functioning within the context of forced displacement.

## Methods

### Setting

Approximately 304,269 Sri Lankan Tamil refugees have been displaced in India since the onset of the Sri Lankan civil war in the 1980s, continuing until 2012 [54]. As of August 2022, the Indian government reports that there are a total of 58,506 Sri Lankan refugees residing in 108 refugee camps in Tamil Nadu—a state in southern India with cultural affinities to the ethnic Tamil refugees [54,55]. An additional 34,135 refugees are also displaced in Tamil Nadu, with many financially resourced urban Tamil refugees opting for non-camp settings. The flow of refugees into India over the years have been influenced by various factors, including India's changing refugee policies and with the de/escalation of violent conflicts in Sri Lanka [56].

The present study was conducted in partnership with the Organization for Eelam Refugee Rehabilitation, or OfERR, a non-profit group founded in 1984. The organization's mission is to help and provide resources to Sri Lankan refugees in India by working to ensure sustainable development inside refugee camps. Participants were recruited at a refugee camp located in Trichy, a city in India's southern state of Tamil Nadu. As one of several 'special' camps initially set up to house Sri Lankan refugees deemed high security threats [57], the camp houses about 1500 families. Recruitment and data collection at the Trichy camp lasted approximately one year, between June of 2014 to August of 2015. The institutional review board at the authors' university approved all practiced study procedures (IRB HM20000475).

### Sampling

The third and seventh authors, along with a team of 12 trained researchers—all female health workers and current residents at Trichy refugee camp–conducted interviews among 120 parent-adolescent dyads (120 parents and 120 adolescents) from the Trichy refugee camp. The research data collection team presented information about the study, such as inclusion criteria and notes about voluntary participation, at community events hosted by OfERR for the Sri Lankan refugee community. Stratified purposive sampling techniques were used to select participants in the study. Inclusion criteria for the study were 1.) Sri Lankan refugee status; 2.) Tamil ethnicity; 3.) participation of one parent-adolescent dyad in a family; and 4.) adolescents must have been between the ages of 13 and 18 at the time of data collection. A specific process was used to ensure a representative sample of 40 families from each subdivision of the refugee camp. First, a number between 1 and 1500 was randomly generated to establish a starting point. Next, every 8 th household was selected for the study. The research team visited 180 homes, after which 120 families, with children and parents living together, contacted them to inform the team of their interest in participating in the study.

### Data collection

Based on the individual preference of the potential participant, the research team described the study and performed the screening process using phone calls in the participants' preferred language (Tamil or English). All participants were proficient in at least one of these languages. Selected participants were then notified to arrange a time and date to sign the Tamil-translated

consent form and partake in a face-to-face interview. All participating parents provided informed consent for themselves and for their children to participate in the study and voluntary adolescent participation was verified during screening by confirming with both the parent and adolescent of the dyads. To deal with potential ethical issues concerning informed consent from adolescents, the research team thoroughly explained the study in simple language to ensure comprehension of data collection methods and use. Moreover, the research team actively assessed participant comprehension and encouraged participants to only disclose what they felt comfortable sharing. Adolescents were not required to answer questions they did not want to respond to. In addition to ensuring informed consent, the research team conducted MINI exams to assess participants' mental and cognitive status. When more than one adolescent in the same age group from one family volunteered to participate, the research team assessed the child's MINI exam and coordinated with parents and health workers to select one child based on need.

Refugees who have experienced negative interactions with authorities and/or foreign cultural traditions may be suspicious of written consent forms. The option of oral consent was provided in these cases, and clear procedures on how to obtain and record verbal consent were given to the research team. Verbal consent was always gained from adolescents. In situations in which gatekeepers had been involved in participant recruitment, it was necessary that research respondents understood their right to refuse participation at any stage in the research process. This refusal would not affect available services or level of care provided. The anonymity of the participants was treated with extreme consideration, given that an agency/service had referred them. The only job of the research team was to collect data, which was then organized, with each parent and adolescent participant receiving assigned codes. Demographic information from the participants was stored at the referral agency and was accessible only to the primary researcher. The research team was instructed on the importance of building relationships with camp residents, building rapport and establishing trust. The team made sure that the relationships were equally important to adolescents. The relationships were aided by previous collaboration between researchers and camp residents on a few pilot studies in the Trichy camp.

These prior research experiences helped the refugees verify the research team's true intentions and trust the absence of coercion and power imbalance. In the current study, rapport was maintained through research and regular attendance at community events sponsored by OfERR; these visits also acted as data collection. Researchers were also able to spend time with adolescents and parents outside of official data collection procedures, getting to know them on a deeper level and engaging with their interests and experiences. This approach is demonstrably useful, as evidenced by the excitement adolescents almost always showed when they saw the first author. The research team was also able to obtain social, linguistic, and facial interaction knowledge with refugee families at the Trichy camp, which helped unwanted interactions remain avoided. Regardless of these precautions and attempts, it is acknowledged that this process may not always reflect autonomous agreement to participate in facets of the study.

At the convenience of the research participants, all interviews were conducted in the OfERR office. Parents and adolescents were interviewed separately in different rooms. When it came to interviews between researchers and adolescents, the research team was eager to ensure that the adolescents were not only comfortable speaking to them, but also that the adolescents were able to see them as neither a 'teacher' nor a 'parent', to guarantee the adolescents would feel safe sharing their experiences. This detail was particularly important given that young children typically provide socially desirable and/or confirmatory responses to please adults. If a male was being interviewed, an elderly woman stood a few feet away from the door to ensure the researchers' safety, as is custom in the participants' tradition. After the

conclusion of an interview, participants engaged in a debriefing session with the researcher, in which they could discuss questions or concerns. After the interview, every family was compensated 1000 Indian

Rupees for their participation, given to the parents as per custom. Each interview lasted approximately two hours with three five-minute breaks.

## Measurement

**Child depressive symptoms.** The child mental health variable consisted of one variable: child depressive symptoms. Child depressive symptoms were measured using the 27-item Child Depression Inventory (CDI) [58]. As one of the most widely used and best studied scales in measuring child depressive symptoms [59,60], the CDI captures affective, cognitive and behavioral symptomatology consistent with depression. Participants were asked to indicate items that describe them best in the past two weeks. Scores were summed for a total score of depression. Scores range from 0–54 with higher scores indicating higher depression. Although cutoff scores for both clinical and nonclinical settings have been suggested, they are culture-bound, and thus not used in this study. This decision was made in due part to the cultural norms with this study population and for the purpose of this research, which is to identify relationships among the key study variables and not focus on identifying depression. Cronbach's alpha for this sample was .772, indicating good reliability.

**Parental mental health.** Parental mental health was measured using the Brief Symptoms Inventory Scale (BSI) [61]. The BSI consists of a total of nine dimensions, however, this study utilized four domains including depressive symptoms (six items), hostility (five items), anxiety symptoms (six items), and somatization (seven items). Using a 5-point Likert Scale ranging from 1 (not at all) to 5 (extremely), participants were asked how often they experienced an item in the last seven days. Responses were then summed to arrive at a total score, with higher scores indicating higher levels of mental distress. BSI scores ranged from 0–79, with an average score of 27.3. Reliability was assessed for each subscale. Cronbach's alpha was calculated at .700 for the somatization subscale, .818 for the depressive symptoms, .776 for anxiety symptoms, and .664 for hostility subscales.

**Child daily stressors.** Child daily stressors were assessed using the 29 item Children's Daily Stressors Scale (CDSS) [62]. This scale was specifically developed for Sri Lankan youth who have been exposed to war and trauma. The purpose of the scale is to examine the contributions of war, disaster exposure and daily stressors on the mental health of children [62]. It includes three subscales: depression (ten items), inter-parental conflict (4 items), and abuse (6 items). On a 3-point Likert Scale ranging from 1–3, participants were asked how stressful they currently found each item (such as inadequate water/housing, physical abuse by authority figures, and parental substance use) to be. Responses included "very stressful", "somewhat stressful", and "not stressful". Higher scores indicated higher levels of daily stress. Cronbach's alpha was calculated to be .848, indicating high reliability.

**Family functioning.** Two variables comprise the family functioning variable: Quality of Parenting and Family Cohesion. Quality of parenting was measured using a shortened version of the Child Report of Parent Behavior Inventory (CRPBI). Specifically, the 20-item Acceptance Subscale [63,64] was used to assess children's perceptions of parental acceptance and quality of caregiving. Adolescent participants were asked a series of parenting-specific behaviors (e.g., "understands your problems and your worries", "enjoys doing things with you") to rate as either "A lot Like", "Somewhat like", or "Not like" their parents. Higher total scores of parental quality signify lower levels of parental quality. Reliability for this scale was calculated at .843. Family Cohesion was assessed using a shortened, 18-item version of the Child Self-

Report of Family Inventory Scale (SFI) [65]. The full, 36-item SFI assesses family competency and comprises six domains, including Health/Competence, Conflict, Cohesion, Leadership, Communication and Emotional Expressiveness [66]. The Communication and Cohesion domains were used in this study. Cronbach alpha for this sample was .864, indicating high reliability.

**Child sleep quality.** To assess the presence of sleep problems among the adolescent sample, researchers administered the Pittsburgh Sleep Quality Index. The scale includes a total of 15 items pertaining to respondent's sleeping patterns in the past two weeks. Cronbach's alpha for this sample was calculated at .811, indicating high reliability. Higher total scores indicate poor sleep quality.

**Child total strengths and difficulties.** To assess adolescent behavioral and emotional problems, the Child Report of Total Strengths and Difficulties Questionnaire (SDQ) [67] was used. The SDQ is widely used across the globe for children between the ages of 11 and 17 [68]. The scale comprises a total of 25 items and five subscales including emotional, conduct problems, hyperactivity/inattention, peer relationship problems and prosocial behaviors. Cronbach's alpha was .746 for this sample, indicating good reliability.

## Statistical analysis

To test the associations and predict the variables contributing to child depressive symptoms, we use both linear and nonlinear methods, namely, hierarchical regression and XGBOOST. We used XGBOOST to demonstrate the power of this method in data analysis. Utilizing these methods in conjunction allows us to test the variable importance and help overcome limitations presented by the data. While linear models allow us to test model significance, Mean Square Errors, adjusted $R^2$, and change in $R^2$ ($R^2\Delta$), they are limited in model flexibility and explaining variable importance when predicting the outcome variable. Thus, we adopted a supplemental method with the ability to explain variable importance of the predictors and perform concurrent cross validation. Using XGBoost, we illustrate which variables are important in predicting child depressive symptoms, and also examine the dependency of each predictor. For testing the significance of the regression slope ($\beta = 0$ versus $\beta$ not equals 0) through two-sided tests, a sample size of n = 117 yields an 80% power to detect a somewhat small effect size of 0.25. In the case of a medium effect size of 0.3, the test with n = 117 achieves a power of 92%, while for a large effect size of 0.5, the aforementioned test with n = 117 exhibits a power of 99.9%. For a detailed explanation on effect sizes [69]. All power computations were performed using G-Power v3.1 software. Power analysis for XGBoost models could not be conducted due to the absence of a linear, parametric model form.

**Linear model.** SPSS version 28 was used to conduct bivariate and multivariate (specifically hierarchical regression) analyses. We conducted a hierarchical regression analysis to test predictors for child depressive symptoms. In step one, child specific variables were entered into the model (child sleep quality, child strengths and difficulties and child daily stressors). In step two, parent-specific mental health variables were entered into the model (parent mental health). In the final step, family functioning variables (parenting quality and family cohesion) were added to the model.

**Extreme Gradient Boosting (XBOOST).** In addition to the linear model, we implemented a nonlinear machine learning algorithm, namely Extreme Gradient Boosting (XGBoost). XGBOOST provides a flexible framework for modeling child depressive symptoms using the six predictor variables. The XGBoost algorithm is a regularized tree boosting method originally proposed by Chen & Guestrin [70]. This method has achieved excellent predictive performance in many fields, and is considered extremely suitable for the statistical analysis of

big data [71]. The predictive model for child depressive symptoms was fitted using the XGboost method and the variable importance of the six predictors were calculated concurrently. Lastly, SHapley Additive exPlanations (SHAP) summary plots, proposed by Lundberg and colleagues [72] were used to visually interpret the impact of these predictors on the response. The analysis was done in R software using "SHAPforxgboost" package [73].

**XGBoost model setup.** The data analysis was performed on open-source RStudio version 4.2.0 using packages "xgboost," "caret," "mlr," "ROSE," "DMwR," and "ggplot2". We split the dataset into 80% training data, and 20% testing data. The model was fitted on the training dataset, and then fine-tuned through the boosting procedure using a k-fold cross-validation technique until convergence. For details on the algorithm, see Chen and Guestrin [70].

**SHapley Additive exPlanations (SHAP) method.** The Shapley value, coined by Shapley [74,75], is a game theoretic approach of assigning credit to 'players' in a fair manner depending on their contribution to the total output of a 'game'. Quantifying Shapley values for features in regression or machine learning models is a recent development, first implemented by Lipovetsky and Conklin [76], Cohen and colleagues [69], and later developed by Štrumbelj and Kononenko [77]. This work led to the development of the SHAP package for Python [78]. Nowadays, Shapley values are widely used in applied machine learning areas such as medicine [79,80], chemistry and pharmaceutical research [81,82], economics [83], and environmental research [84].

Shapley values quantify the contribution of each predictor variable to the prediction, as compared to a baseline average prediction. For linear models, these values are just the values of each feature multiplied by their corresponding weights. However, the quantifications are complex for non-linear models. To connect the game-theoretic context to model based predictions, consider the 'game' as a prediction task. The corresponding 'gain' is the actual prediction after including a particular predictor minus the model's average prediction, and the 'players' are the predictors that collaborate to receive the gain (i.e., predict a certain value). For technical details on Shapley values, including mathematical formulation, see Lundbeg and colleagues [85]. For further information on the XGBoost and Shapley Additive Explanations (SHAP) methods, please refer to S1 Text in the supporting information section.

## Results

### Sample characteristics and descriptive statistics

Descriptive statistics for all key variables are presented in Table 1. Parents were overwhelmingly female (96%) and married (87%). Average age for parents was 38 years old. Average number of children per parent was about three (2.86), and average time spent in the camp was 22.67 years. Average age of child participants was 14.23 years, with average score for child depressive symptoms at 11.61 ($SD$ = 5.54). Mean scores for total parent BSI scores was 27.28 (SD = 6.4) and ranged from 0 to 79.; average scores for child daily stressors was 18.42 ($SD$ = 16.34); child depressive symptoms mean score was 11.6 (SD = 5.5); average child sleep quality was 5.81 (SD = 5.21). Mean levels of parenting quality was 48.41 (SD = 13.10) and family cohesion at 55.98 ($SD$ = 13.21).

### Linear analysis

The data met the assumptions for regression analysis. The residuals were independent, and there was no concern for multicollinearity, as supported by the correlation matrix table and the VIF scores for each variable (not shown here). Additionally, the variables were all linear with the outcome variable, and the assumption for homoscedasticity was met. Hierarchical regression analysis was conducted to examine the main effects of parental mental health, child-specific variables, and family functioning variables on child mental health. Results from

**Table 1. Descriptive statistics: Demographic & key variables.**

| Descriptive statistics: Demographic & Key Variables | | | | | | | |
|---|---|---|---|---|---|---|---|
| Frequency (%) | M | SD | Mode | Range | Median | Missing | |
| *Demographics* | | | | | | | |
| **Parent** | | | | | | | |
| Male | 4 (3.4) | | | | | | |
| Female | 115 (95.8) | | | | | | |
| Age | – | 38.00 | 6.62 | 35 | 36 | 37 | |
| Leng.in camp | – | 22.67 | 4.98 | 24.00 | 31.00 | 24.00 | |
| Married | 100 (87) | | | | | | |
| Widowed | 10 (8.3) | | | | | | |
| Separated | 5 (4.2) | | | | | | |
| # Children | – | 2.86 | .963 | 3.00 | 5 | 3.00 | |
| Employed | 38 (31.7) | | | | | | |
| Unemployed | 79 (65.8) | | | | | | |
| **Child** | | | | | | | |
| Age | | 14.23 | 1.64 | 13.00 | 6.00 | 14.00 | |
| Sex | | | | | | | 1 |
| Male | 43(37%) | | | | | | |
| Female | 73(63%) | | | | | | |
| *Mental Health* | | | | | | | |
| Child Depress | | 11.6 | 15.5 | | 26 | 11 | 2 |
| Parent BSI | | 27.3 | 18.5 | | 79 (0–79) | 25 | |
| Child Daily stress | | 18.4 | 16.3 | | 55(0–55) | 11.7 | 1 |
| Total Difficulties | | 10.2 | 5.4 | | 22 (2–24) | 10 | |
| *Family Functioning* | | | | | | | |
| Parenting Quality 48.1 7.6 33.7(26.3–60) 50 | | | | | | | |
| Family cohesion 56.0 13.3 52.3(26.5–78.8) 57.3 | | | | | | | |
| *Child Sleep Quality* 5.8 5.2 26(0–26) 5 | | | | | | | 2 |
| *n = 117* | | | | | | | |

the regression analysis, including standardized regression coefficients (β), $R^2$ and $R^2\Delta$ are summed in Table 2. Results from bivariate correlation test and previous literature on child mental health were used to determine the variables included in the model. In the first model, 29.8% of the variance in child depressive symptoms was accounted for, $R = .546$, $F(3, 107) = 15.127$, $p < .001$. In the second model, 32.8% of the variance in the response variable was accounted for, $R = .573$, $F(4, 106) = 12.939$, $p < .001$. In the last model, 34.6% of the variance in child depressive symptoms was accounted for by the variables included in the model, $R = .588$, $F(6, 104) = 9.177$, $p < .001$.

Results (Table 2) indicated two statistically significant predictors of child depressive symptoms in the model while keeping other predictors fixed, we see that every unit of increase in parent mental health predicted a .171 increase in child depressive symptoms ($p = .037$). Additionally, every unit increase in child total difficulty and strengths predicted a .441 increase in child depressive symptoms ($p < .001$). Parenting quality was not significant (p = .097).

## XGBOOST results

Only seven iterations were needed to obtain the final model (with the smallest mean square error), and this final model was used to understand the variable importance of the predictors.

**Table 2. Hierarchical regression results.**

*Hierarchical Regression Results*

| | B | 95% CI | SE | β | $R^2$ | Δ $R^2$ |
|---|---|---|---|---|---|---|
| Constant | 5.678*** | [3.529, 7.83] | 1.084 | | .298 | |
| Child Sleep Qual | -.022 | [-.202, .158] | .091 | -.021 | | |
| Child daily stress | .009 | [-.044, .061] | .026 | .028 | | |
| Diff & Strength | .550*** | [.376, .723] | .088 | .553 | | |
| Constant | 4.526*** | [2.170, 6.883] | 1.188 | | .328 | .030 |
| Child Sleep Qual | -.008 | [-.186, .169] | .090 | -.008 | | |
| Child daily stress | .009 | [-.043, .060] | .026 | .027 | | |
| Diff & Strength | .516*** | [.343, .690] | .087 | .519 | | |
| Parent BSI | .053** | [.005, .101] | .024 | .177 | | |
| Constant | 11.285* | [2.567, 20.004] | 4.396 | | .346 | .018 |
| Child Sleep Qual | -.033 | [-.212, .147] | .091 | -.032 | | |
| Child daily stress | .005 | [-.050, .059] | .028 | .014 | | |
| Diff & Strength | .438*** | [.240, .636] | .100 | .441 | | |
| Parent BSI | .051** | [.003, .099] | .024 | .171 | | |
| Parenting Quality | .000 | [-.258, .022] | .035 | .001 | | |
| Family Cohesion | -.117* | [-.069, .070] | .070 | -.163 | | |

* $p < .05$

** $p < .01$

*** $p < .001$

Below (Table 3) is the result detailing the variable importance of the model with six predictors. Note that these six predictors were also used for the linear model analysis.

Variable importance shows the relative contribution of each predictor to model accuracy and is measured by 'Gain'. In Table 3, each feature and their corresponding Gain scores are presented. Gain values are similar to the % of variance explained by each predictor in the model. In our analysis, child strength and difficulties was the most important variable in predicting child depressive symptoms with the highest variable importance of about 43%. Child sleep quality also contributed significantly to the child's depressive symptoms since the Gain value (about 24%) was second highest, followed by child daily stressors score with Gain value of about 13%. In addition, family cohesion and parenting quality each contributed about 8% to the prediction accuracy of child depressive symptoms.

Finally, the SHAP summary plot is used to explain the effect of each individual predictor on the response. Note that high Shapley values means that the effect was severe and low Shapley means that the effect is not severe. Fig 1 shows the SHAP summary plot that explains the effect of the six predictors on the response (child depressive symptoms).

**Table 3. Variable importance for the fitted XGBoost model.**

| Feature | Gain |
|---|---|
| Child Strength & Difficulties | 43% |
| Sleep Quality | 24% |
| Child Daily Stressors | 13% |
| Parenting Quality | 8% |
| Family Cohesion | 8% |
| Parent Mental Health | 3% |

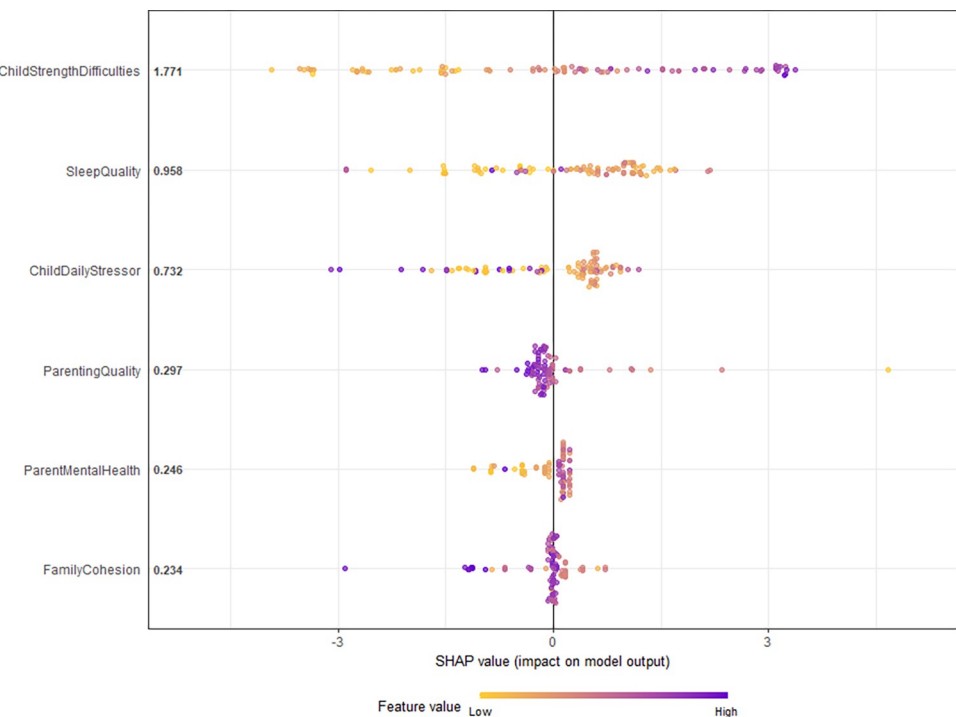

**Fig 1. SHAP values (impact on model output).**

Fig 1 shows that the child total strength and difficulties scores has a high positive effect on child depressive symptoms, therefore child depressive symptoms would increase as the child total strength and difficulties scores increased. Furthermore, parenting quality, sleep quality, child daily stressor and family cohesion have moderately high positive effects on child depressive symptoms. This means that depressive symptoms in children could worsen if parents have negative behavior towards them, if the family cohesion is poor or if the children do not have good sleep. Lastly, parent mental health had a moderate positive effect on the child's depressive symptoms, thus the child's depressive symptoms would moderately increase if parents had poor mental health. Also, note that the values in Fig 1 are not the variable importance percentages, but rather the means associated with each variable and hence are different from the values shown in Table 3.

## Discussion

The combined results from the linear and nonlinear analyses provides predictive models for child depressive symptoms and identifies parental mental health, child total strength and difficulties and family functioning variables as key predictors in child depressive sysmptoms. These results are promising as they begin to explore potential factors associated with the mental health of refugee children. As demonstrated in prior research, the nature of living in a refugee camp are individually stressful for both parent and child, and collectively in family systems. This stress is often accompanied by a lack of material resources, due, in part, to the abrupt nature of forced migration that compels families to flee without first securing essential resources [86].

In addition to forced displacement, Sri Lankan refugee families have witnessed traumatic and violent events and have undergone a great degree of loss [87]. Refugees often come to

refugee camps with high levels of pre-migration trauma [31] that devastates mental and physical wellbeing and adaptive coping. High levels of premigration stress become proliferated in a context characterized by a hostile socio-legal environments undergirded by Indian refugee policies. As one of the most marginalized categories of refugees, Sri Lankan camp refugees contend with discrimination and poverty in their host environments, and must navigate the challenges of precarious conditions resulting from prolonged displacement.

Upon arrival to the camps, refugees must prepare to begin the process of living in displacement for an undetermined period of time and grapple with the traumatic and daily stressors associated with life in protracted displacement. Such challenges include disruptions in protective and adequate parenting styles and family structures that are often crucial in cultural child rearing practices, and the emergence of intergenerational conflicts in the context of liminality. Such experiences have great potential to disrupt the mental health of most people, let alone families with both extensive collective and individual trauma histories, and a battered resource reserve. It may be appealing to conclude that given these circumstances, we are limited in influencing the adverse exposure to distressing experiences for youth experiencing transmigration. However, findings of this study underscore the vital role of interventions to address child psychological distress (including social and emotional problems) inside refugee camps and to improve family functioning.

Previous literature on parenting inside refugee camps have underscored the need for parenting resources inside refugee camps [88,89]. In addition to grappling with psychological adjustments, parents must also learn how to navigate complex relationships with their children in distressing environments. As parents learn to adapt to new environments, their ability to respond to their children may also shift [90]. Prior to displacement, families may have had the financial, material, cultural and personal resources to carry out parenting roles effectively, which become depleted or deteriorated after initial displacement occurs. Further, refugee camps are often overcrowded, under-resourced, and unprepared to respond to the psychosocial needs of the families contained in them. For families faced with daily stressors, they may not have the capacity or luxury to fully engage in reflective parenting practices [91].

Our nonlinear model also revealed a significant association between sleep quality and child depressive symptoms. This makes intuitive sense, since when a child is depressed, sleep quality often becomes compromised. Even though this effect is moderate and complex to quantify, a possible intervention to help improve sleep in this population may be worthwhile.

There are a few notable limitations in this study. First, the measurements used may not be accurate in fully capturing the constructs under study in this paper. Specifically, only the Children's Daily Stressors scale was developed for use with Sri Lankan populations. Even then, the initial scale was developed with a sample of youth not experiencing transmigration, nor externally displaced [61]. Next, there may be another construct that may better explain the mental health outcomes of youth experiencing transmigration, namely a transmigration stress construct. Identifying the specific stressors associated with displacement and contributing to poorer mental health for children is important for this population. Displaced refugees in camps may be more vulnerable to encounter traumatic stressors than perhaps other refugees who have either resettled or have not left their home country [92]. The acculturative stress, coupled with transmigration stress may contribute to mental health uniquely, and without proper measurement tools, it is unlikely that we may know the unique experiences of displaced refugee adolescents. This challenge of assessing specific issues with this population also leads to the difficulty of explaining in detail the variations seen in both models.

Despite the extensive literature detailing the risk for psychopathology among refugee youth, an important caveat in these discourses revolves around culturally-bound measurement tools and the (re)construction of health and illness in various cultural contexts. With few

studies validating measurement tools in refugee populations, it is important to consider how constructs of resilience and distress interface with both language and the cultural meanings embedded in experiences. In this regard, qualitative studies assessing the mental health experiences of refugee youth may offer valuable insights. Qualitative studies on refugee camp children corroborate trends found in quantitative literature, namely detailing the traumatic nature of refugee camps [33,93], and emphasize the need to culturally-ground our understanding of the resilience and wellbeing process and adopt a structured approach [93–95]. Future research should adopt qualitative designs to continue to develop valid measurement tools, as well as to gain an understanding of cultural constructs, such as health, wellbeing, and parent-child relationships.

In summary, this study begins to theorize the relationships among parental mental health, family functioning and child depressive symptoms in the context of forced displacement. Findings offer evidence that within the context of transmigration, there is a notable deterioration in family functioning, leading to declines in both parental and child mental health outcomes. Findings underscore the need to develop and implement resources that build upon parenting capacities and that improve parental efficacy. Other key implications include the urgency to provide mental health interventions for both parents and their children in camp settings, and to immediately address adverse transmigration stressors.

## Supporting information

**S1 Text. XGBoost and Shapley Additive Explanations (SHAP) methods.**
(DOC)

## Acknowledgments

We thank the reviewers for their comments for improving the manuscript.

## Author Contributions

**Conceptualization:** Muna Saleh, Miriam Kuttikat, Indranil Sahoo, David Chan.

**Data curation:** Miriam Kuttikat, Hannah George.

**Formal analysis:** Muna Saleh, Elizabeth Amona, Indranil Sahoo, David Chan, Kyeongmo Kim.

**Funding acquisition:** Miriam Kuttikat.

**Investigation:** Miriam Kuttikat, Indranil Sahoo.

**Methodology:** Miriam Kuttikat.

**Project administration:** Miriam Kuttikat, Hannah George.

**Resources:** Elizabeth Amona, Kyeongmo Kim.

**Software:** Muna Saleh, Elizabeth Amona, Indranil Sahoo.

**Supervision:** Miriam Kuttikat, Indranil Sahoo, David Chan, Kyeongmo Kim, Hannah George.

**Validation:** Muna Saleh, Elizabeth Amona, Kyeongmo Kim.

**Visualization:** Muna Saleh, Elizabeth Amona, Indranil Sahoo, David Chan.

**Writing – original draft:** Muna Saleh, Elizabeth Amona, Jennifer Murphy, Marianne Lund.

**Writing – review & editing:** Miriam Kuttikat, Jennifer Murphy, Hannah George, Marianne Lund.

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
