## [Decision Letter · Decision Letter 0]

6 Dec 2023

PONE-D-23-22224Child Mental Health Predictors Among Camp Refugees: Utilizing Linear and XGBOOST ModelsPLOS ONE

Dear Dr. Kuttikat,

Thank you for submitting your manuscript to PLOS ONE. After careful consideration, we feel that it has merit but does not fully meet PLOS ONE’s publication criteria as it currently stands. Therefore, we invite you to submit a revised version of the manuscript that addresses the points raised during the review process.

We look forward to receiving your revised manuscript.

Kind regards,

Kamrul Hsan, MS

Academic Editor

PLOS ONE

Journal Requirements:

"This work was supported by National institute of Health Fogarty International [grant numberK01 TW 009648]."

"No"

Additionally, because some of your funding information pertains to [commercial funding//patents], we ask you to provide an updated Competing Interests statement, declaring all sources of commercial funding.

In your Competing Interests statement, please confirm that your commercial funding does not alter your adherence to PLOS ONE Editorial policies and criteria by including the following statement: "This does not alter our adherence to PLOS ONE policies on sharing data and materials.” as detailed online in our guide for authors http://journals.plos.org/plosone/s/competing-interests.  If this statement is not true and your adherence to PLOS policies on sharing data and materials is altered, please explain how.

Please include the updated Competing Interests Statement and Funding Statement in your cover letter. We will change the online submission form on your behalf.

Reviewers' comments:

Reviewer's Responses to Questions

**Comments to the Author**

1. Is the manuscript technically sound, and do the data support the conclusions?

Reviewer #1: Partly

Reviewer #2: Yes

2. Has the statistical analysis been performed appropriately and rigorously? 

Reviewer #1: Yes

Reviewer #2: Yes

3. Have the authors made all data underlying the findings in their manuscript fully available?

Reviewer #1: Yes

Reviewer #2: Yes

4. Is the manuscript presented in an intelligible fashion and written in standard English?

Reviewer #1: No

Reviewer #2: Yes

5. Review Comments to the Author

Reviewer #1: Child mental Health Predictors among the Camp Refugees

My general comment: The manuscript has great potential to generate good scientific results in the area of traumatic stress and refugee mental health studies. However, the study’s objectives are not clear, a little confusing and hard to follow. Can the authors come up with hypotheses too.

Please see my brief specific comments below.

Abstract:

Objectives

I fail to understand the distinction between the following objectives

1.Describe the best predictors of Children’s depressive symptoms

2. Determine the variable importance of predictors of depressive ……

3. To provide “initial” what do the authors mean by “Initial”

-Please provide the statistical software used in the analysis

Generally, the abstract is confusing: The authors should bear in mind the structure of the abstract as follows.

a) Background/ introduction

b) Objective

c) Methods

d) Results

e) Conclusions and Recommendations

The Background to the study is well written but brief. The authors could benefit from the following previous studies 1: https://pubmed.ncbi.nlm.nih.gov/24478246/Journal of traumatic stress, 27(1), 35–41. https://doi.org/10.1002/jts.2189 (2) https://doi.org/10.1080/20008198.2017.1283086https://www.ncbi.nlm.nih.gov/pmc/articles/PMC5328389/

It is not clear whether the authors are assessing the children on their personal experiences or the parent’s potentially traumatic experiences.

Methods are described and written well.

Results: Have significant potential. However, since the objectives weren’t clear from the inception, the flow of the results poses difficulty to readability.

Reviewer #2: The manuscript was technically good and sound. But there were some issues to be addressed and amendments. Please se the details at the PDF. Issues were given as comments on the PDF.

Why only 120 samples were measured? Is there any limitation? Does it exceeds minimal sample size? What is the number of adolescents of that camp? (See all comments at the manuscript PDF file)

You can upload the data to any public repository or give with the manuscript as supplementary file.

6. PLOS authors have the option to publish the peer review history of their article (what does this mean?). If published, this will include your full peer review and any attached files.

Reviewer #1: No

Reviewer #2: **Yes: **Abu Bakkar Siddique

---

## [Author Response · Author response to Decision Letter 0]

26 Feb 2024

Journal Requirements:

Response: We have followed the format suggested. One reviewer asked us to deviate from the abstract format which we have ignored to stay with the suggested format requested by the editor.

Response: Thank you for this suggestion. We do plan on putting the data in a repository once we have completed the analysis of the data for future papers. We have multiple papers analyzing this data currently, and will make the data available once we have completed these manuscripts.

"This work was supported by National institute of Health Fogarty International [grant numberK01 TW 009648]."

"No"

Additionally, because some of your funding information pertains to [commercial funding//patents], we ask you to provide an updated Competing Interests statement, declaring all sources of commercial funding.

In your Competing Interests statement, please confirm that your commercial funding does not alter your adherence to PLOS ONE Editorial policies and criteria by including the following statement: "This does not alter our adherence to PLOS ONE policies on sharing data and materials.” as detailed online in our guide for authors http://journals.plos.org/plosone/s/competing-interests. If this statement is not true and your adherence to PLOS policies on sharing data and materials is altered, please explain how.

Response: The authors declare this statement is true

Please include the updated Competing Interests Statement and Funding Statement in your cover letter. We will change the online submission form on your behalf.

Response: This has been completed. 

Response: We have added the IRB number to the manuscript (bottom of page 9). We provide a discussion on our ethical considerations on page 10 to 11. 

Reviewers' comments:

Reviewer's Responses to Questions

Comments to the Author

Reviewer #1: Child mental Health Predictors among the Camp Refugees

My general comment: The manuscript has great potential to generate good scientific results in the area of traumatic stress and refugee mental health studies. However, the study’s objectives are not clear, a little confusing and hard to follow. Can the authors come up with hypotheses too.

Response: All changes to the manuscript are highlighted in yellow. We have added a “Present Study” section in the manuscript to better illustrate the goals and objectives of the study. We have added three central research questions to this section and have chosen not to include a hypothesis. We hope these modifications are sufficient. 

Please see my brief specific comments below.

Abstract:

Objectives

I fail to understand the distinction between the following objectives

1.Describe the best predictors of Children’s depressive symptoms

2. Determine the variable importance of predictors of depressive ……

3. To provide “initial” what do the authors mean by “Initial”

-Please provide the statistical software used in the analysis

Response: This has been addressed. Note that the statistical software was already mentioned in the manuscript.

Generally, the abstract is confusing: The authors should bear in mind the structure of the abstract as follows.

a) Background/ introduction

b) Objective

c) Methods

d) Results

e) Conclusions and Recommendations

Response: Thank you for the feedback. We have revised our abstract for clarity, however, we have chosen to remain with an unstructured/narrative abstract. We note that The journal allows for both abstract formats.

The Background to the study is well written but brief. The authors could benefit from the following previous studies 1: https://pubmed.ncbi.nlm.nih.gov/24478246/Journal of traumatic stress, 27(1), 35–41. https://doi.org/10.1002/jts.2189 (2) https://doi.org/10.1080/20008198.2017.1283086https://www.ncbi.nlm.nih.gov/pmc/articles/PMC5328389/

Response: We have significantly expanded on our background and have provided much needed contextual information for the study. We hope these changes will be welcomed by the reviewers. 

It is not clear whether the authors are assessing the children on their personal experiences or the parent’s potentially traumatic experiences.

Response: Thank you for this feedback! It is difficult to completely separate these two factors. In general, both of these factors play an important role in a child's mental health and we consider both. We have added on to the background section to better illustrate our central rationale undergirding the study, including the addition of a Present Study section. We hope these additions make our objectives more clear. 

Methods are described and written well.

Results: Have significant potential. However, since the objectives weren’t clear from the inception, the flow of the results poses difficulty to readability.

Response: We have rewritten the objectives/hypotheses.

Reviewer #2: The manuscript was technically good and sound. But there were some issues to be addressed and amendments. Please see the details at the PDF. Issues were given as comments on the PDF.

Response: We have modified as the reviewer asked with a few exceptions. The style of abstract is different from the format given by the format suggested by the style template so no change was made. This is also the case with the limitations and recommendations. The comment in the introduction asking about the importance of the study is clearly stated by the lines highlighted in this section. Another comment asked about OfERR which the reviewer perhaps missed, as it was already included in the manuscript. The question at the beginning of the Data Collection section asked whether the seven year old data is represented for today, which we feel is not an appropriate question since this manuscript is focused on the analysis of the collected data, and not on the predictive validity of the data. There is another comment in this section that asks about the fraction which gave oral or written consent; the authors feel that this is irrelevant when it comes to the analysis, as both forms of consent are valid. There is a question on the reasons for only taking 4 male participants, but there is no reason for this aside from convenience sampling. This includes all the data collected. 

Why only 120 samples were measured? Is there any limitation? Does it exceeds minimal sample size? What is the number of adolescents of that camp? (See all comments at the manuscript PDF file)

Response: Thank you for the feedback. We have included a power analysis (page 16) that indicates the sample size is acceptable. We hope this addition addresses questions related to sample size.

---

## [Editor Report · Decision Letter 1]

30 Apr 2024

Child Mental Health Predictors Among Camp Refugees: Utilizing Linear and XGBOOST Models

PONE-D-23-22224R1

Dear Dr. Kuttikat,

We’re pleased to inform you that your manuscript has been judged scientifically suitable for publication and will be formally accepted for publication once it meets all outstanding technical requirements.

Kind regards,

Kamrul Hsan, MS

Academic Editor

PLOS ONE

---

## [Editor Report · Acceptance letter]

2 Jul 2024

PONE-D-23-22224R1 

PLOS ONE

Dear Dr. Kuttikat, 

I'm pleased to inform you that your manuscript has been deemed suitable for publication in PLOS ONE. Congratulations! Your manuscript is now being handed over to our production team.

Kind regards, 

on behalf of

Dr. Kamrul Hsan 

Academic Editor

PLOS ONE